# Myanmar’s Land Cover Change and Its Driving Factors during 2000–2020

**DOI:** 10.3390/ijerph20032409

**Published:** 2023-01-29

**Authors:** Yiming Wang, Yunfeng Hu, Xiaoyu Niu, Huimin Yan, Lin Zhen

**Affiliations:** 1State Key Laboratory of Resources and Environmental Information System, Institute of Geographic Sciences and Natural Resources Research, Chinese Academy of Sciences, Beijing 100101, China; 2School of Geosciences, Yangtze University, Wuhan 430100, China; 3College of Resources and Environment, University of Chinese Academy of Sciences, Beijing 100049, China

**Keywords:** land mapping spatial distribution, dynamic change, driving factor, national-scale analysis

## Abstract

Land use/cover change (LUCC) research occupies an important place in the study of global change. It is important for the ecological protection and long-term development of a place. Current research is lacking in the study of dynamic changes at the national level in Myanmar over long time periods and sequences. Quantitative research on the driving factors of LUCC is also lacking. This paper uses the GLC_FCS30 (Global Land-Cover product with Fine Classification System) dataset and socio-economic statistical data in Myanmar to conduct the study. The dynamic change process of LUC (land use/cover) was investigated using the land use dynamic degree, land use transfer matrix, and Sankey diagram. Principal component analysis was used to derive the main drivers of LUCC. The drivers were quantified using multiple linear stepwise regression analysis and specific factors were analyzed. The spatial scope of the study is Myanmar, and the temporal scope is 2000–2020. Results: (1) In 2020, the spatial distribution of LUC in Myanmar shows predominantly forests and croplands. Forests account for 56.64% of the country’s total area. Agricultural land accounts for 25.59% of the country’s total area. (2) Over the time scale of the study, the trend of LUCC in Myanmar showed significant shrinkage of evergreen broad-leaved forest and deciduous broad-leaved forest (a total shrinkage of −3.34 × 10^4^ km^2^) and expansion of the other land types. (3) Over the time scale of the study, the dynamic changes in LUCC in Myanmar most occurred as an interconversion between two land types, such as between cropland and deciduous broad-leaved forest, evergreen broad-leaved forest and shrubland, deciduous broad-leaved forest and shrubland, evergreen broad-leaved forest and evergreen needle-leaved forest, and evergreen broad-leaved forest and deciduous broad-leaved forest. (4) The dynamics of LUC in Myanmar is mainly influenced by the socio-economic level of the country. Among them, the impact of agricultural level is the most obvious. Specifically, Myanmar’s LUCC is mainly driven by urban population, urbanization rate, industrial value added, food production, and total population. Our research will enable the Myanmar government to make more scientific and rational land management and planning and to make more informed decisions. After understanding the basic situation of LUCC in Myanmar, the hydrological effects, biodiversity changes, and ecological service function changes due to land change in the region can be explored. This is the direction of future research.

## 1. Introduction

LUCC studies are indispensable in global change research. LUCC is capable of exhibiting climate change characteristics. The natural ecosystems on the Earth’s land surface can be affected by human activities. This impact can also be manifested through LUCC. Therefore, LUCC plays the role of a bridge and a link in the process of interaction between humans and nature [1]. As early as 1995, in order to increase understanding of the dynamic processes of LUCC and study their relationship with total environmental change of the Earth, IGBP and IHDP established the LUCC program. In 2006, IGBP and IHDP initiated the Global Land Project, which took over and replaced the LUCC program. Its main purpose is to synthesize and organize the ideas, knowledge, and methods from the research conducted by scientists in the land systems science community [2]. In 2014, ICSU and ISSC launched the 10-year Future Earth Initiative in collaboration with several international organizations such as UNESCO and UNEP. The program can provide society with the critical knowledge it needs to respond to the problems posed by environmental change on all of Earth’s surfaces. At the same time, it can provide the world with the opportunity to make the transition to sustainable development. It was the result of the gradual succession of global environmental change research strategies such as LUCC and GLP and met the core needs of the international community to emphasize linking land science research with addressing social challenges [3].

The basis of LUCC research is the LULC (Land Use & Land Cover) dataset. In the 1980s, Wilson and Henderson-Sellers of the University of Liverpool [4] produced a global land use/cover dataset for use in GCM climate models. After entering the 21st century, remote sensing observation technology and land mapping technology have developed rapidly. As a result, more and more LULC dataset products with global scale, higher spatial resolution, and shorter update cycles have appeared in the world, such as the IGBP DISCover product of the US Geological Survey [5] (https://daac.ornl.gov/, accessed on 30 September 2022), the GLC product of the EU Joint Research Center [6] (https://forobs.jrc.ec.europa.eu/, accessed on 29 September 2022), and the European Space Agency’s GLOBCOVER product [7,8] (https://www.esa.int/, accessed on 30 September 2022). These LULC products are mostly land mapping products with a spatial resolution of 300 m, 500 m, or even 1 km, and an update cycle of 5 years or even 10 years. In the past ten years, with the emergence of higher temporal and spatial resolution earth observation platforms and the accumulation of historical observation data, especially the further development of remote sensing cloud technology, higher spatial resolution (30 m, even 10 m), and higher time, the resolution (annual scale) LULC dataset was produced. Other datasets include the Globeland30 dataset produced by the Ministry of Natural Resources of China [9] (http://www.globallandcover.com/, accessed on 30 September 2022), the GLC_FCS30 dataset produced by the Academy of Aerospace Technology, Chinese Academy of Sciences [10,11,12] (https://data.casearth.cn/, accessed on 30 September 2022), the LSV10 data product produced by the European Space Agency [13] (https://esa-worldcover.org/, accessed on 30 September 2022), and ESRI 2020 Land Cover 10 m (ESRI10) data product produced by ESRI Corporation of the United States [14] (https://www.arcgis.com/, accessed on 30 September 2022).

Currently, the global LULC is changing rapidly under the influence of both the natural environment and human activities. This may break the balance of ecosystems and reduce biodiversity [15]. Therefore, when conducting research on LUCC, the analysis of how its drivers affect the sustainable development of the region is the focal point. Many studies have shown that the key element driving change in LULC is the economic activity of humans in society. Liu et al. [16] analyzed LUCC in China during the 1990s. Cropland area increased the most (44.88 million mu). The grassland area decreased the most (51.55 million mu). Policy regulation and economic drivers are the main causes of LUCC and its spatial and temporal variation. Moisa et al. [17] studied the LUCC in Addis Ababa municipality in Ethiopia. With the increase in population, vegetation cover and other land cover types have been transformed into building land. From 1991 to 2021, the agricultural land cover decreased by 66.8 km^2^ and the vegetated land cover decreased by 25.7 km^2^. The built-up area of the city was 96.6 km^2^ (18.3%) in 1991 and increased to 277.2 km^2^ (52.6%) in 2021. The development of urban society was the main reason for LUCC. Hu et al. [18] analyzed the LUCC in Ha Long City, Vietnam, since the founding of the country. The share of forest land shrank by 26.3%. The share of both urban land and industrial and mining storage has expanded by 4.3 times. The main reason for this change was the improvement in socioeconomic level. These studies provide examples that can be used for the extraction, research framework, and indicator methodology of LULC change impact factors for a particular country or region.

In studying land use change, we often use land use dynamic degree to analyze the extent of land type change and the land use transfer matrix to analyze the land type transformation process. These methods were used by Talebi et al. [19], Zhao et al. [20], and Xing et al. [21] in their study of land use dynamic degree in the east Meskine region of Ardabil province, Manwan Dam of the Lancang River, and low elevation areas of Ethiopia. When performing the driving mechanism of LUCC, there are generally multiple driving factors. The advantage of performing analysis with multiple drivers is that it makes the analysis results more comprehensive. The disadvantages are that it will generate data redundancy and a computationally intensive and complex analysis process. Moreover, there may be a correlation between multiple drivers. This may lead to less reliable final analysis results. Jiang et al. [22] and Liu et al. [23] used both principal component analysis and linear regression models in their study of the driving factors of LUCC in the Cuno River Basin and Chinese Jiangxi province.

Myanmar has long suffered from instability, sectarian violence, corruption, weak infrastructure, and a long history of colonial exploitation. The country’s economic and cultural level is generally relatively backward. According to a study by the United Nations Development Programme, Myanmar is extremely backward in human development in 2020, ranking 147th out of 189 countries in the world [24]. Previously, LULC studies related to Myanmar were usually conducted at smaller regional scales (such as watersheds, protected areas, counties, cities, and other administrative regions). For example, the study by McGinn et al. [25] studied LUCC in the Chindwin River Basin, Myanmar. During 1999–2019, the forest area in this area decreased by about 2%. The mining area increased by 0.38%. The area of agricultural land decreased by 2.1% overall. The increase in deforestation and mining activities has led to major problems such as water pollution, sedimentation, and river channel changes. Zaehringer et al. [26] studied LUCC in southeastern Myanmar from 1990 to 2017 and showed that the area shifted from smallholder rotation to natural rubber, betel nut, cashew, and oil palm trees. From this, it could be concluded that commercial interests were the core force for the huge loss of forest resources. The building land area increased from 1988 (730.64 ha) to 2018 (3607.20 ha). The urban forest area decreased from 2008 (4901.63 ha) to 2018 (3558.36 ha).

Although there are many studies on LUCC in Myanmar, there are still many problems. Among the many LULC data products that exist, the classification of various land types has mixed results. For example, ESRI has released the 2020 Land Cover product. It has good classification results for agricultural land, forests, water bodies, and building sites, but poor classification results for shrubs, wetlands, or fallow lands [27]. Additionally, Myanmar is relatively backward. Its socio-economic data and climate data are inadequate. This has prevented scholars from conducting fine-grained LUCC process portrayal and pattern description studies on it. For the analysis of the driving factors, most of the studies in Myanmar also stay at the stage of qualitative analysis. There are fewer quantitative analyses. Helen et al. [28] from the University of Tokyo, Japan, analyzed the LUCC process in the Myanmar city of Pyin Oo Lwin between 1988 and 2018. It was largely driven by demographic and socio-economic factors. This study provides evidence for urban green space protection, green infrastructure development planning, etc., but the study does not quantify the influencing factors. Moreover, the spatial scope of LUCC research is not large, and the time scale is also short. In addition, there is also a lack of in-depth discussions on the analysis of driving mechanisms based on mathematical statistics for national sustainable development and ecological protection.

In this paper, we use the long time series, high-resolution GLC_FCS30 dataset as the LUC study data. We use climate and national socio-economic statistics as the driving factor research data. The dynamic processes of spatial and land type change of LUC in Myanmar were analyzed using land use dynamic degree, Sankey diagram, and land use transfer matrix. Principal component analysis was applied to derive the main drivers of LUCC in Myanmar. Multiple linear stepwise regression in statistics was used to quantify the drivers of LUCC in Myanmar and to analyze the specific drivers. This study can enable the Myanmar government to conduct land planning and designing in a more scientific and practical way. Additionally, it can enable Myanmar to achieve sustainable development faster and protect the ecological environment better.

## 2. Data and Methods

### 2.1. Study Area Overview

Myanmar is in the northwestern part of the Indochina Peninsula (9°58′–28°31′ N, 92°20′–101°11′ E) (Figure 1). Myanmar is bordered by China in the north. It is connected to Laos and Thailand in the east. It is bordered to the southwest by the Andaman Sea and the Bay of Bengal in the Indian Ocean. It is bordered by Bangladesh and India to the northwest. The total land area is 67.66 × 10^4^ km^2^, and the coastline is 0.28 × 10^4^ km. Myanmar has high terrain in the north and low terrain in the south. It is surrounded by mountain ranges in the north, west, and east. It has high mountainous areas in the north (the Savage Mountains, also known as the Kachin Mountains, the Kumaon Ridge, and the Hukawng Valley Mountains). The Rakhine Mountains and the Naga Hills are in its western part. The Shan State plateau is to the east of it. Khyakabo Peak, near the Chinese border, is 5881 meters above sea level and is the highest peak in Myanmar. The Irrawaddy alluvial plain lies between its western mountains and eastern plateau. The terrain is low and flat, with an average elevation of 200 meters to 600 meters.

Myanmar is a tropical monsoon climate region. Its temperature does not vary much throughout the year. The whole year of Myanmar can be divided into three seasons: hot season, rainy season, and cool season. The hot season lasts from March to mid-May. The rainy season is from mid-May to October. The cool season is from November to February. The coldest month is January. It has an average temperature of 20–25 °C. The hottest months are April and May. It has an average temperature of 25–30 °C. Myanmar receives abundant rainfall. The most rainfall occurs in June, July, and August. The southwest monsoon is prevalent during this period. Precipitation in Myanmar is followed by May, September, and October. The rainfall in various places from May to October accounts for about 90–95% of the annual rainfall. Most of Myanmar has an annual rainfall of 4000 mm or more. The central part of Myanmar (Magway) is a rain shadow area with relatively low annual rainfall, but it also reaches 1000 mm.

Myanmar is divided into 7 provinces, 7 states, and 2 centrally administered municipalities. The seven provinces are Yangon, Mandalay, Bago, Magway, Sagaing, Ayeyarwady, and Tanintharyi. The seven states are Chin State, Shan State, Kayah State, Mon State, Kachin State, Karen State, and Rakhine; the two municipalities are Naypyidaw and Yangon. The province is the main inhabited area of the Burmese, and the state is the inhabited area of various ethnic minorities. Before 2005, the capital of Myanmar was Yangon, which is located in the southwest and borders the Andaman Sea. In 2005, the Myanmar government moved the capital to Naypyidaw in the central part of the Sittang Valley between the Bago Mountains and the Bonnong Mountains.

The social and economic level of Myanmar is relatively backward. The industry and service industries are underdeveloped. It is an agricultural country. The rural population accounts for about 70% of the country’s total population. Rural people make their living mainly from agricultural farming. The main crops grown include conventional rice, wheat, corn, and peas. It also grows a large number of cash crops, such as rubber, sugarcane, cotton, and palm. In recent years, peas have surpassed rice as the most important agricultural product exported by Myanmar. Myanmar is rich in forestry resources. Forests cover more than 56% of the country’s total land area. Myanmar produces and exports a large amount of teak, and the output of teak ranks first in the world. Myanmar produces and exports large quantities of teak. The production of teak ranks first in the world. In Myanmar’s industry, the labor-intensive processing and manufacturing industry represented by the textile and garment industry and some light industry sectors are developing fast. The development of all kinds of heavy industries is seriously lagging behind.

### 2.2. Data Sources

We downloaded the GLC_FCS30 dataset for the study from the China Earth Big Data Science Project Data Sharing Service (https://data.casearth.cn/, accessed on 30 October 2022). The spatial resolution of this dataset is 30 meters. It covers the period 1985–2020. Among them, every 5 years is 1 time period. It has a total of 8 time periods [12]. There are two levels of land cover types. Among them, there are 9 first-level types and 30 second-level types (Table 1), with classification accuracies of 82.5% and 68.7%, respectively [10]. There are 22 second-class land types in Myanmar (Table 2). In Myanmar, forests cover more than half of the total area of the country. We would like to study it in more detail and conclude the distribution and changes of each specific type of forest in Myanmar. So, only forests used Tier 2 land types in the study. All other land types used Tier 1 land types. The areas of bare areas and permanent snow and ice were extremely small. We merged bare areas into grasslands. We merged permanent snow and ice into wetlands & water bodies.

We calculated meteorological data (X1, X2) for Myanmar for 2000–2020 from day-by-day data from GLDAS-2.1 [29] and the PERSIANN-CDR dataset [30]. Their spatial resolution is 0.25 radians. We downloaded data on socio-economic development (X3–X9) from the World Bank (https://www.shihang.org/zh/home, accessed on 1 November 2022). We downloaded the rest of the data (X10–X16) from the World Food and Agriculture Organization (http://www.fao.org/, accessed on 3 November 2022).

Wood production includes wood fuel, sawn wood, veneer logs, veneer sheets, plywood, and other products. Vegetable production includes the production of green garlic, peppers, onions and shallots, other vegetables, etc. Oil crop production includes the production of sesame seeds, groundnuts, sunflower seeds, castor oil seeds, mustard seeds, cotton seeds, cashew nuts, etc., of which groundnuts production accounts for about 50%. Food production includes the production of wheat, soya beans, sorghum, rice, potatoes, millet, maize, lentils, peas, beans, cassava, sweet potatoes, cereals, etc., among which rice accounts for about 80%. Fruit production includes the production of areca nuts, coconuts, sugar cane, bananas, mangoes, guavas, mangosteens, other fruits, etc., of which sugar cane accounts for about 75%.

By analyzing various statistics of Myanmar, we established a system of driving factors of LUCC in Myanmar (Table 3).

### 2.3. LUCC Analysis Methods

Generally, there are two types of land use dynamic degree. One is the integrated land use dynamic degree. The other is the single land use dynamic degree.

As the name implies, the integrated land use dynamic degree (*D*) provides an integrated representation of the overall degree of change in all land use types within a study area [31]. It is able to express the quantitative characteristics of the variation. The mathematical expression is:(1)$$D=\left\{\overset{n}{\underset{i=1}{\sum }}\left(\frac{\Delta U_{i}}{U_{i}}\right)\right\}\times \frac{1}{T}\times 100\%$$

In the expression, D is the integrated land use dynamic degree in the research period; $\Delta U_{i}$ is the area of change of class *i* LUC throughout the study; $U_{i}$ is the area of class *i* LUC at the beginning of the study; T is the research period; and *n* is the number of LUC types.

The single land use dynamic degree (*K*) can be used to quantify the changes in the number of a particular land type within a certain time frame in different subdivisions of the study area. Therefore, it can compare the changes in the number of land types in different subdivisions, and then predict its development trend [32]. The formula is:(2)$$K=\frac{U_{b}-U_{a}}{U_{a}}\times \frac{1}{T}\times 100\%$$

In the formula, *K* is the dynamic degree of a particular LUC type during the time period of this study; U*_a_* denotes the area of a particular LUC at the start of the study; U*_b_* denotes the area of the same LUC at the end of the study; and T denotes the time period studied.

The Markov model is a statistical model. It has been widely applied to various scientific fields. The land use transfer matrix is built on its basis. The land use transfer matrix allows the transfer between different land types to be quantified [33]. Moreover, the land use transfer matrix has to be used with a specific time interval when it is carried out. The land use transfer matrix can reveal the process by which various land types transform into each other during this time interval [34]. The land use transfer matrix is expressed mathematically as follows:(3)$$S_{ij}=\left[\begin{array}{cccc} S_{11}\; & S_{12} & \cdots & S_{1n}\\ S_{21} & S_{22} & \ldots & S_{2n}\\ \vdots & \vdots & \vdots & \vdots \\ S_{n1} & S_{n2} & \cdots & S_{nn} \end{array}\right]$$
where S denotes the area of various types of land; *i* denotes the LUC type at the beginning of the study; *j* denotes the LUC type at the end of the study; and *n* denotes the number of LUC types.

We obtained areas of all land types within Myanmar by using the GLC_FCS30 dataset from 2000 to 2020. We calculated two land use dynamic degrees in Excel by Equations (1) and (2). We imported the land use data of Myanmar from 2000 to 2020 in the GLC_FCS30 dataset into ArcGIS 10.2. The land use transfer matrix was obtained by raster calculation.

### 2.4. Driving Factors Analysis Methods

Principal component analysis (PCA) uses an orthogonal transformation. It transforms the original random variable with correlated components into a new variable with uncorrelated components. Algebraically speaking, it converts the covariance array of the original variables into a diagonal array. Geometrically speaking, it transforms the original system of variables. After this operation, it makes it a new orthogonal system. In the case of a scattered distribution of sample points, it refers to the most open orthogonal direction of the scattering. In turn, the dimensionality reduction of the multidimensional variable system is achieved [35]. It converts multiple indicators into several comprehensive indicators. At the same time, it retains as much data and information in it as possible. Additionally, it is also able to eliminate redundancy arising from multiple data [36]. Through the principal component analysis method, the extracted principal component factors can be given new names according to the unique meanings reflected by professional knowledge and indicators. Thus, reasonable explanatory variables are obtained [37]. The selection method for each main factor is as follows:(4)$$\{\begin{array}{c} F_{1}=c_{11}Z_{1}+c_{12}Z_{2}+\cdots +c_{1n}Z_{n}\\ F_{2}=c_{21}Z_{1}+c_{22}Z_{2}+\cdots +c_{2n}Z_{n}\\ \vdots \\ F_{n}=c_{n1}Z_{1}+c_{n2}Z_{2}+\cdots +c_{nn}Z_{n} \end{array}$$
where *F_i_* denotes principal component *i*, *i* = 1, 2, …, *n*; *c* denotes the eigenvector corresponding to the eigenvalues of the covariance matrix; and *Z* denotes the value obtained after normalization of the original variables. In this formula, for each *i*, there is $c_{i1}^{2}+c_{i2}^{2}+\cdots +c_{in}^{2}=1$.

There are multiple drivers that affect LUCC. We use a multiple linear stepwise regression approach to conduct the analysis. In this method, variables are introduced in and out. It introduces variables individually. When each of the variables is entered, the variables that have been selected for inclusion are checked individually. When the previously entered variables become no longer significant due to the later introduced variables, they are removed. This ensures that only variables with significant effects are included in the regression model before each new variable is added. The introduction and elimination of variables was repeated in this manner. This operation is performed until neither influential significant independent variables are entered into the regression model nor influential insignificant independent variables are removed from the regression model [38]. Multiple linear stepwise regression analysis methods can build the most optimum or appropriate regression model. This allows a more in-depth study of the dependencies between variables [39]. This method can be used to make predictions for the dependent variable. It can improve the accuracy of the results. We also used this method to conduct the analysis of each driving factor presented in the paper. The expressions are as follows:(5)$$Y=\beta +\alpha_{1}\;X_{1}+\alpha_{2}\;X_{2}+\ldots +\alpha_{n}X_{n}$$
where α_1_, α_2_, …, α_n_ denote relevant coefficients; β denotes the constancy term; Y denotes the dependent variable; and $X_{1}$−$X_{n}$ denote the independent variables.

We imported Myanmar land use data and driving indicator data from 2000 to 2020 into SPSS software. After setting the relevant parameters, we let SPSS perform the corresponding operations. The final results of principal component analysis and multiple linear stepwise regression analysis were obtained. From the results, we can see the main driving forces and specific drivers of LUCC in Myanmar from 2000 to 2020.

## 3. Results Analysis

### 3.1. Spatial Distribution

In 2020, it can be seen from the statistical map of various types of LUC areas in Myanmar (Figure 2) that Myanmar is dominated by forests (accounting for 56.64% of the area) and cropland (accounting for 25.59% of the area). The combined area of the two land types accounts for more than 82% of the total national land area.

Specifically, forests have the widest distribution of area (43.18 × 10^4^ km^2^, 56.64%). Among them, most are evergreen broad-leaved forests (19.06 × 10^4^ km^2^, 25.01%) and deciduous broad-leaved forests (18.00 × 10^4^ km^2^, 23.62%), and a few are evergreen needle-leaved forests (6.11 × 10^4^ km^2^, 8.02%). The second largest LUC type is cropland (19.51 × 10^4^ km^2^, 25.59%). The third largest LUC type is shrubland (10.96 × 10^4^ km^2^, 14.37%). The area of grassland, impervious surfaces, wetlands and water bodies, etc., is small (total 2.59 × 10^4^ km^2^, 3.39%). They contain grassland (0.48 × 10^4^ km^2^, 0.63%), impervious surfaces (0.48 × 10^4^ km^2^, 0.64%), and wetlands and water body (1.62 × 10^4^ km^2^, 2.13%).

Cropland is mostly located in the central Irrawaddy alluvial plain, such as Sagaing, Magway, Mandalay, etc. There is also a small amount of distribution in the southwestern coastal areas, such as Ayeyarwady, Yangon, Mon State, etc.

Forests are mostly located in the Shan State plateau in the east, the northern region, and the mountainous region in the west. Among them, the evergreen broad-leaved forest is mostly located in Kachin State in the northern mountain area and Tanintharyi, Bago, and other places in the south. There is also a small amount of distribution in Chin State, Magway, Rakhine, and other places in the mountainous and hilly area in the west and Shan State area in the eastern Shan State plateau. The deciduous broad-leaved forest is mostly located in the alluvial plains of the central Ayeyarwady River, such as Bago, Sagaing, Magway, etc., and there is also a small amount of distribution in the Shan State area of the eastern Shan State plateau. The evergreen needle-leaved forest is mostly located in Kachin State in the northern mountain region, and the rest are scattered in the western mountain region and the eastern Shan State plateau.

In addition, the shrubland is mostly located in the Rakhine and Chin State areas in the western mountains, and the rest are scattered in the eastern Shan State plateau, northern mountainous areas, southern Mon State, Karen State, Tanintharyi, and other places. The grasslands are scattered in Kachin State, Sagaing, and other places in the northern mountains, and in the Shan State area of the eastern Shan State plateau. Impervious surfaces are concentrated in large cities, such as Sagaing and Mandalay in the central part, Yangon and Moulmein in the southern part, and others are distributed in the capitals or larger cities of various provinces or states. Wetlands and water bodies are generally distributed in points in Kachin State and Sagaing in the northern mountains, Shan State in the eastern Shan State plateau, Rakhine in the western mountains, and Bago, Yangon, Ayeyarwady, Tanintharyi, and other places in the south.

### 3.2. Analysis of Land Dynamic Change

During 2000–2020, the area of forest in Myanmar has shrunk (−3.34 × 10^4^ km^2^, −0.36%), and the area of other land types has expanded. Among them, deciduous broad-leaved forest (−2.43 × 10^4^ km^2^, −0.59%) and evergreen broad-leaved forest (−1.86 × 10^4^ km^2^, −0.44%) showed a decreasing trend, and evergreen needle-leaved forest (+0.94 × 10^4^ km^2^, +0.91%) showed an increasing trend. Cropland (+0.98 × 10^4^ km^2^, +0.26%), wetlands and water bodies (+0.24 × 10^4^ km^2^, +0.85%), and impervious surfaces (+0.94 × 10^4^ km^2^, +0.91%) all showed a trend of area expansion. The most expanded area is shrubland (+1.82 × 10^4^ km^2^, +1.00%), and the least expanded area is grassland (+0.09 × 10^4^ km^2^, +1.19%).

In terms of the spatial distribution of land use dynamic degree, the main LUC changes in the first ten years (2000–2010) occurred in the central region, and the LUC changes in the eastern and western regions were weaker. In the next ten years, the LUC changes were most concentrated in the western and southern regions, and then in the eastern region, the intensity of LUC changes also climbed rapidly (Figure 3).

During 2000–2005, the spatial distribution of integrated land use dynamic degree in Myanmar was high in the central plains, low in the eastern plateaus and western mountains, and basically followed the law of decreasing with the elevation of the terrain. The integrated land use dynamic degree in most provinces is below 33%. Among them, the lowest is Rakhine (10.87%), and the highest is Yangon (56.06%).

During 2005–2010, the integrated land use dynamic degree increased in the mountains of western Myanmar and the plains of central Myanmar and decreased in southern Tanintharyi. The integrated land use dynamic degree in half of the provinces is above 33%. Among them, the lowest is Shan State (8.79%). The highest is Mon State (2156.12%), followed by Karen State (755.98%).

During 2010–2015, the integrated land use dynamic degree in Myanmar declined as a whole, and most provinces were below 33%. Among them, the lowest is Sagaing (9.98%). The highest is Yangon (1141.52%), followed by Kayah State (260.41%).

During 2015–2020, the integrated land use dynamic degree in Myanmar increased as a whole, showing a trend of increasing with the elevation of the terrain. Only Mandalay (19.09%) is below 20%. There are four provinces with a dynamic degree exceeding 100%, the highest being Chin State (1269.01%), followed by Kayah State (730.66%), Shan State (208.89%), and Rakhine (105.38%).

### 3.3. Source and Destination

During 2000–2020, the land area where transfer between different LUC types occurred in Myanmar was 18.51 × 10^4^ km^2^. From 2000 to 2010, the land area where LUC types were transferred in Myanmar was 14.39 × 10^4^ km^2^, and from 2010 to 2020, the area of transferred land was 16.84 × 10^4^ km^2^. The conversion process mainly occurs between two land use/cover types (Figure 4), including cropland and deciduous broad-leaved forest, evergreen broad-leaved forest and shrubland, deciduous broad-leaved forest and shrubland, evergreen broad-leaved forest and evergreen needle-leaved forest, and evergreen broad-leaved forest and deciduous broad-leaved forest.

The cropland continues to show a trend of expansion. During 2000–2020, converted cropland to other land was a total of 2.60 × 10^4^ km^2^, mainly converted into the deciduous broad-leaved forest (−55.82%, −1.45 × 10^4^ km^2^), wetlands and water bodies (−15.29%, −0.40 × 10^4^ km^2^), and shrubland (−12.95%, −0.34 × 10^4^ km^2^). The total area of other land converted to cropland is +3.58 × 10^4^ km^2^, mainly from the deciduous broad-leaved forest (+67.13%, +2.40 × 10^4^ km^2^) and shrubland (+13.99%, +0.50 × 10^4^ km^2^). Cropland shrinkage mainly occurs in the central plains such as Mandalay, Magway, Ayeyarwady, and Yangon; cropland expansion mainly occurs in the northern mountainous Sagaing, Kachin State, etc., the southern coastal Karen State, Mon State, Tanintharyi, etc., and the eastern plateau Shan State (Figure 5a).

The overall forest showed a shrinking trend. During 2000–2020, the total area of forest converted to other land was 7.16 × 10^4^ km^2^. The total area of other land converted to forest was 3.40 × 10^4^ km^2^.

Among them, the total area of evergreen broad-leaved forest transformed into other land types was 4.72 × 10^4^ km^2^, mainly transformed into shrubland (−48.10%, −2.27 × 10^4^ km^2^), evergreen needle-leaved forest (−26.36%, −1.24 × 10^4^ km^2^), and deciduous broad-leaved forest (−19.36%, −0.91 × 10^4^ km^2^). The total area of other land converted to the evergreen broad-leaved forest was 2.84 × 10^4^ km^2^, mainly from shrubland (+37.51%, +1.06 × 10^4^ km^2^), deciduous broad-leaved forest (+32.16%, +0.91 × 10^4^ km^2^), and evergreen needle-leaved forest (+27.25%, +0.77 × 10^4^ km^2^). The shrinking evergreen broad-leaved forests are mainly distributed in Kachin State, Sagaing, Rakhine, etc., in the northern and western mountains and Tanintharyi, Karen State, etc., in the south. The expansion is mainly distributed in Chin State, etc., in the western mountainous areas, and Shan State, etc., in the eastern plateau.

The total area of deciduous broad-leaved forest transformed into other land types was 6.35 × 10^4^ km^2^, mainly transformed into cropland (−37.83%, −2.40 × 10^4^ km^2^), shrubland (−28.37%, −1.80 × 10^4^ km^2^), and evergreen needle-leaved forest (−16.93%, −1.08 × 10^4^ km^2^). The total area of other land converted to the deciduous broad-leaved forest was 3.91×10^4^ km^2^, mainly from cropland (+37.02%, +1.45 × 10^4^ km^2^), shrubland (+24.58%, +0.96 × 10^4^ km^2^), evergreen broad-leaved forest (+23.34%, +0.91 × 10^4^ km^2^), and evergreen needle-leaved forest (+13.90%, +0.54 × 10^4^ km^2^). The shrinkage of deciduous broad-leaved forests mainly occurred in Shan State and Kayah State in the eastern plateau, Sagaing and Chin State in the northern and western mountains, and Bago, Mon State, and Karen State in the south-central region. The expansion mainly occurred in Magway in the central plain, in the northern Sagaing and Kachin State in the mountains, Shan State in the eastern plateau, etc.

The total area of evergreen needle-leaved forest transformed into other land types was 1.55 × 10^4^ km^2^, mainly transformed into the evergreen broad-leaved forest (−49.93%, −0.77 × 10^4^ km^2^) and deciduous broad-leaved forest (−35.14%, −0.54 × 10^4^ km^2^). The total area of other land converted to the evergreen needle-leaved forest was 2.51 × 10^4^ km^2^, mainly from the evergreen broad-leaved forest (+49.62%, +1.24 × 10^4^ km^2^) and deciduous broad-leaved forest (+42.91%, +1.08 × 10^4^ km^2^). The shrinkage of evergreen coniferous forests mainly occurred in the Kachin State in the northern mountainous area and the Shan State in the eastern plateau. The expansion mainly occurred in the Chin State in the western mountainous area and the Shan State in the eastern plateau (Figure 5b).

The shrubland showed a continuous expansion trend. During 2000–2020, the total area of shrubland converted to other land was 2.68 × 10^4^ km^2^, mainly transformed into the evergreen broad-leaved forest (−39.74%, −1.06 × 10^4^ km^2^), deciduous broad-leaved forest (−35.93%, −0.96 × 10^4^ km^2^), and cropland (−18.72%, −0.50 × 10^4^ km^2^). The total area of other land transformed into shrubland was 4.52 × 10^4^ km^2^, mainly from the evergreen broad-leaved forest (+50.25%, +2.27 × 10^4^ km^2^) and deciduous broad-leaved forest (+39.91%, +1.80 × 10^4^ km^2^). The shrinkage of the shrubland has a small distribution in the central plains of Mandalay, Mandalay, and other areas. Other areas are scattered. Shrubland expansion mainly occurred in the eastern plateau, western and northern mountains, and southern coastal areas (Figure 5c).

### 3.4. Climate Change and Socio-Economic Development

During 2000–2020, the overall climate in Myanmar showed a slightly warm and dry trend (Figure 6a). Over the past 20 years, the average annual temperature for many years has been 27.04 °C, showing an overall upward trend (+0.09 °C/10a), reaching a peak in 2019 (28.26 °C). The average annual total precipitation for many years is 1799.93 mm, showing a fluctuating decline overall trend (−118.8 mm/10a). The total annual precipitation in 2017 was the least (1333.98 mm).

From 2000 to 2020, the overall population of Myanmar showed a continuous increase trend (Figure 6b). The total population of Myanmar increased from 2000 (4671.97 × 10^4^) to 2020 (5440.98 × 10^4^). These years have increased by 769.01 × 10^4^. It grew at an annual rate of 7.6‰. The rural population increased from 2000 (3409.37 × 10^4^) to 2020 (3746.60 × 10^4^). These years have increased by 337.23 × 10^4^. It grew at an annual rate of 4.7‰. The urban population increased from 2000 (1262.6 × 10^4^) to 2020 (1694.38 × 10^4^). These years have increased by 431.78 × 10^4^. It grew at an annual rate of 1.48%. The rate of urbanization continues to accelerate from 2000 (27.03%) to 2020 (31.14%). These years have increased by 4.12 percentage points.

From 2000 to 2020, the overall economic development of Myanmar showed an upward trend (Figure 6c). GDP increased from 2000 (USD 68.49 billion) to 2020 (USD 789.30 billion). These years have increased by USD 720.81 billion. It grew at an annual rate of 13.00%. The agricultural value added increased from 2000 (USD 35.54 billion) to 2020 (USD 152.65 billion). These years have increased by USD 117.10 billion. It grew at an annual rate of 7.56%. The industrial value added increased from 2000 (USD 6.35 billion) to 2020 (USD 229.14 billion). These years have increased by USD 222.79 billion. It grew at an annual rate of 19.64%.

From 2000 to 2020, the overall forestry development in Myanmar showed an upward trend (Figure 6d). Except for a slight decline in natural rubber production after 2018, it rose steadily in other years and reached its peak in 2018 (27.55 × 10^4^ t). Wood charcoal production fluctuated slightly during the rising process, reaching peaks (43.82 × 10^4^ t) in 2019 and 2020. Wood production increased rapidly before 2005 and from 2010 to 2013, showed a downward trend after 2013, and reached a peak in 2013 (4690.84 × 10^4^ m^3^).

From 2000 to 2020, the overall development of agriculture in Myanmar showed an increasing trend (Figure 6e). Vegetable production declined slightly in 2012 and then increased year by year. In 2016, it began to decline slowly and fluctuated, maintaining at about 490 × 10^4^ t, and reached a peak in 2015 (522.25 × 10^4^ t). Oil crop production continued to rise before 2010, then fluctuated slowly, and reached a peak in 2010 (330.57 × 10^4^ t). Food production continued to rise before 2010, then declined, and reached a peak in 2010 (4061.62 × 10^4^ t). The overall fruit production is on the rise, from 757.57 × 10^4^ t in 2000 to 1536.36 × 10^4^ t in 2020, an increase of 778.79 × 10^4^ t. It grew at an annual rate of 3.60%.

### 3.5. Analysis of Driving Factors

In Table 3, the magnitudes of the various driving factors we have selected are different. To address this issue, we adopted a standard deviation normalization method for the socio-economic and climate data of Myanmar.

Through principal component analysis, we obtained its component matrix (Table 4). From this we know that from 2000 to 2020, the driving factors of Myanmar’s economy, society, and climate can be divided into three dimensions. F1 can be classified as the socio-economic dimension. It shows an extremely strong positive correlation with eight driving factors. They are total population (X3), rural population (X4), urban population (X5), urbanization rate (X6), industrial value added (X9), wood charcoal production (X11), vegetable production (X13), and fruit production (X16). F2 can be classified as a climate dimension. It shows an extremely strong positive correlation with the total annual precipitation (X2). F3 is classified as the dimension of agriculture (grain planting), which shows a strong negative correlation with food production (X15). In terms of contribution rate, F1 (56.29%) is far greater than F2 (27.14%) and F3 (13.46%). National development (F1) has more impact on LUCC than climate (F2) and food planting (F3).

In the multivariate linear stepwise regression analysis, we exclude the wetland water type. The independent variables are three principal components (F1, F2, and F3). The dependent variable is the area of the remaining seven LUC types. The independent variable and dependent variable are analyzed, respectively, and the regression equation between them is established (Table 5).

From the regression equation, it can be seen that the area of cropland has an obvious positive relationship with socio-economic (F1) and agriculture (staple food cultivation) (F3). It shows that the cropland area has increased while Myanmar’s socio-economic and agricultural development has taken place. The area of evergreen broad-leaved forests and deciduous broad-leaved forests have an obvious negative correlation with socio-economic (F1). It shows that with the development of the social economy in Myanmar, the area of evergreen broad-leaved forests and deciduous broad-leaved forests decreased. The area of shrubland, grassland, and impervious surfaces have a significant positive correlation with socio-economic (F1), respectively. It shows that with the development of society and economy, the area of shrubland, grassland, and impervious surfaces are all increasing. In summary, the main influence on the LUCC situation in Myanmar is socio-economic. The small amount of impact is on agriculture (staple food cultivation). There is no credible statistical relationship between the evergreen needle-leaved forest area and the main components F1, F2, and F3.

Furthermore, based on principal component analysis, we obtained the 10 key driving factors, which were total annual precipitation (X2), total population (X3), rural population (X4), urban population (X5), urbanization rate (X6), industrial value added (X9), wood charcoal production (X11), vegetable production (X13), food production (X15), and fruit production (X16). We used them as independent variables. We used the area of the seven aforementioned LUC types as dependent variables. Then, we established a regression equation using multiple linear stepwise regression methods (Table 6).

From Table 6, we can know that there is a significant positive correlation between the cropland area and the urbanization rate (X6). It has a significant negative correlation with food production (X15). The accelerated rate of urbanization has promoted population growth, and the demand for food, vegetables, and fruits will increase. Due to these markets’ needs, the cropland area will increase. After 2010, while the area of cropland continued to expand, the output of staple food and vegetables continued to decline, but the output of oil crops and fruits continued to increase (Figure 6). This shows that a considerable part of the arable land has been transferred from traditional staple food and vegetable cultivation to high-value-added oil crops and fruit cultivation. Both the area of evergreen broad-leaved forests and deciduous broad-leaved forests showed a significant negative correlation with the urbanization rate (X6). This shows that with the increase in urbanization rate and the construction and expansion of urban land, the evergreen broad-leaved forest and deciduous broad-leaved forest located in the low-lying, suitable for living and development areas are cut down first, reducing their area. The area of shrubland has a significant positive correlation with the urban population (X5). Urbanization increases the size of shrubland by destroying forests and converting them into shrubland. The grassland area has a significant positive correlation with industrial value added (X9). It also has a significant negative correlation with food production (X15). Industrial development will increase the change from forest to shrubland. This will also allow the change of shrubland to grassland. To meet the growing demand for food, it depends on the expansion of the cropland area. The expansion of cropland area is first to occupy the grassland that is easy to cultivate, and then to occupy the forest. There is an obvious positive correlation between the area of impervious surfaces and the total population (X3). The overall population increases, resulting in the need for more places to live. This requires building more houses, building roads, etc. This will increase the area of impervious surfaces. After the analysis, we did not obtain regression equations for the area of evergreen needle-leaved forest and 10 key drivers.

## 4. Discussion

### 4.1. LUCC and Its Influencing Factors

Our research shows that from 2000 to 2020, the area of evergreen broad-leaved forests and deciduous broad-leaved forests in Myanmar decreased. The area of the rest of the land types increased. Among them, the absolute area of shrubland increased the most. This result is the same as the regional-scale research results of Wang et al. [40] in Yangon, Karki et al. [41] in Inle Lake, and McGinn et al. [25] in the Chindwin River Basin. All of their studies also indicate that the forest area decreased, and the areas of shrubland, grassland, impervious surfaces, and wetlands all increased within the study area. The changing trend of LUCC in Myanmar is similar to that of other Indochina countries. Research by Hu et al. [18], Zhang et al. [42], and Niu et al. [43] in Vietnam, Laos, Cambodia, and other countries also found the phenomenon of shrinking forests, shrubland, and the expansion of impervious surfaces.

Studies have shown that 60% of LUCC is related to direct human activities, and the other 40% is indirectly related to climate change [44]. Studies on LULC changes in Vietnam, Cambodia, and Laos also found that national economic development and human activities have a greater impact on LUCC in these countries. Among the factors that have a more pronounced impact are GDP, population, and urbanization rates [42,43,45]. Our research also shows that at the national scale, socio-economic and agricultural development have a greater impact on LUCC in Myanmar. The impact of climate change is smaller.

In terms of forest resources in Myanmar, infrastructure development (such as the construction of China–Myanmar oil and gas pipelines) and logging for commercial purposes have had a significant impact on the country’s forest resources [46,47,48]. Since the 1990s, the Myanmar government has successively promulgated laws and regulations such as “Standards and Indicators for Sustainable Forestry Management”, “Forest Law of the Union of Myanmar”, “Forestry Regulations of the Union of Myanmar”, “Regulations for the Implementation of National Forest Logging”, and other laws and regulations, and from April 1 in 2014, a total ban on the export of logs [49,50]. The protection of forest resources not only depends on the formulation and implementation of laws and regulations but also depends on improving the livelihood of farmers and establishing a sustainable forest industry chain. Therefore, there are two main important tasks for Myanmar in terms of land management and achieving sustainable development. First, Myanmar should make use of the good water and heat conditions provided by the tropical and subtropical climate in which it is located. Secondly, Myanmar should establish a modern forest industry with scientific afforestation, nurturing, logging and high value-added forest products processing industry as its core.

At the same time, the support of the international community for the sustainable development of Myanmar’s national land is also crucial. The IFAD, FAO, and the Myanmar government have reached relevant agreements on these aspects, that is, to strengthen the sustainable management and governance of land, forests, water sources, and ecosystems, and to improve local communities and the ability of farmers to cope with natural and man-made disasters, climate change, and transboundary diseases [51,52]. With the gradual construction and development of the China–Myanmar Economic Corridor planned in the “Belt and Road” initiative, the LUC pattern and evolution process in various parts of Myanmar will be increasingly affected by socio-economic development factors and specific construction projects. How will the land use/cover change in Myanmar and other Indochina Peninsula countries? What impact will the LUCC change have on regional ecological services and farmers’ livelihoods? Can regional sustainability be effectively improved? This will be important content of LUCC research at the national and regional scales in Myanmar in the future.

### 4.2. Uncertainty in Data and Results

A study has been conducted for the GLC_FCS30 data accuracy. After the study, it achieves an overall accuracy of 82.5% [10]. In this dataset, the forest accuracy is 94%, the cropland accuracy is 88%, the shrubland accuracy is 56.8%, the grassland accuracy is 67.3%, the impervious surfaces accuracy is 79.3%, and the water body accuracy is 83.8%. Actually, in most land cover datasets based on remote sensing interpretation, the accuracy of cropland and shrubland is not too high. Cropland and shrubland are easily confused with other types, such as cropland with grassland, shrubland with forest, and shrubland with grassland [53]. The accuracy analysis carried out by Wang et al. [54] based on four sets of LUC datasets based on different satellite remote sensing information sources and different resolutions in the Indochina Peninsula also shows that the degree of confusion of grassland, shrubs, bare land, and construction land is high, and the overall accuracy is low. The impact of the accuracy of the GLC_FCS30 dataset on the results of this paper is inevitable.

The rainfall in various parts of Myanmar from May to October accounts for about 90–95% of the annual precipitation. In different seasons of the year, the range and boundaries of wetlands and water bodies are affected by annual precipitation, and the changes are very large [55]. Therefore, there are great uncertainties in the area statistics and change analysis of wetlands and water bodies in Myanmar in this study. Therefore, we excluded wetlands and water bodies from the LUCC driving factors analysis.

The two driving factor analysis methods used in this study also have limitations. We simply established a linear regression relationship between the values of the driving factors and the values of the LUC areas. However, it does not correspond to it in space. We also did not take into account human factors such as culture, policy, and people’s ideological changes. Therefore, the results of our analysis of driving factors presented so far are coarse resolution in spatial terms. It also does not present spatial differences in the driving factors of LUCC within different provinces, watersheds, and agricultural zones in Myanmar. Moreover, the driving factors presented in this paper are macro-oriented and fail to reflect the influence of Myanmar’s unique Buddhist culture, military government system, national and provincial development plans, and specific construction projects. These shortcomings will depend on the development of a more detailed spatial analysis model of the driving mechanism in the future and the development of multi-scale and cross-scale LUCC driving mechanism analysis cases.

## 5. Conclusions

The data for our study is the GLC_FCS30 dataset and official Myanmar socio-economic statistics. We analyzed the dynamic change characteristics of LUCC in Myanmar from 2000 to 2020. The driving mechanisms of LUCC in Myanmar were summarized. It also suggested possible directions for Myanmar’s forest resource conservation and LUCC research. The present study is a regional reference that can be used for global LUCC studies. It is also the first study of long-time series LUCC processes and driving factors at the national scale in Myanmar.

As a result of our study, it is known that the main LUC types in Myanmar are forests and cropland. During 2000–2020 in Myanmar, forests showed a significant shrinkage trend, and the rest of the land types showed a significant expansion trend. Socio-economic development had a greater impact on LUCC change in Myanmar, and climate change had a smaller impact on LUCC change in Myanmar. We emphasize that in order to protect Myanmar’s disappearing forest resources, the Myanmar government and the international community should work together and take effective measures such as returning farmland to forests, improving national literacy, and developing a green economy.

In the study, we established a method to analyze spatial pattern changes in LUCC. The research basis of this method is the GLC-FCS30 dataset, land use dynamic degree, and Sankey map. We also established an analysis method for the driving factors. The research basis of this method is the principal component analysis and multiple linear stepwise regression analysis. In the future, if scholars want to conduct research on similar LULC changes in other countries or regions, they can learn from and apply these methods. However, the current study is still inadequate in the refinement of the driving mechanism and spatial modeling of the driving mechanism analysis, and more in-depth research is needed in the future. In addition, after understanding the basic situation of land change in Myanmar, the hydrological effects, biodiversity changes, and ecological service function changes due to land change in the region are also directions for future research.

## Figures and Tables

**Figure 1 ijerph-20-02409-f001:**
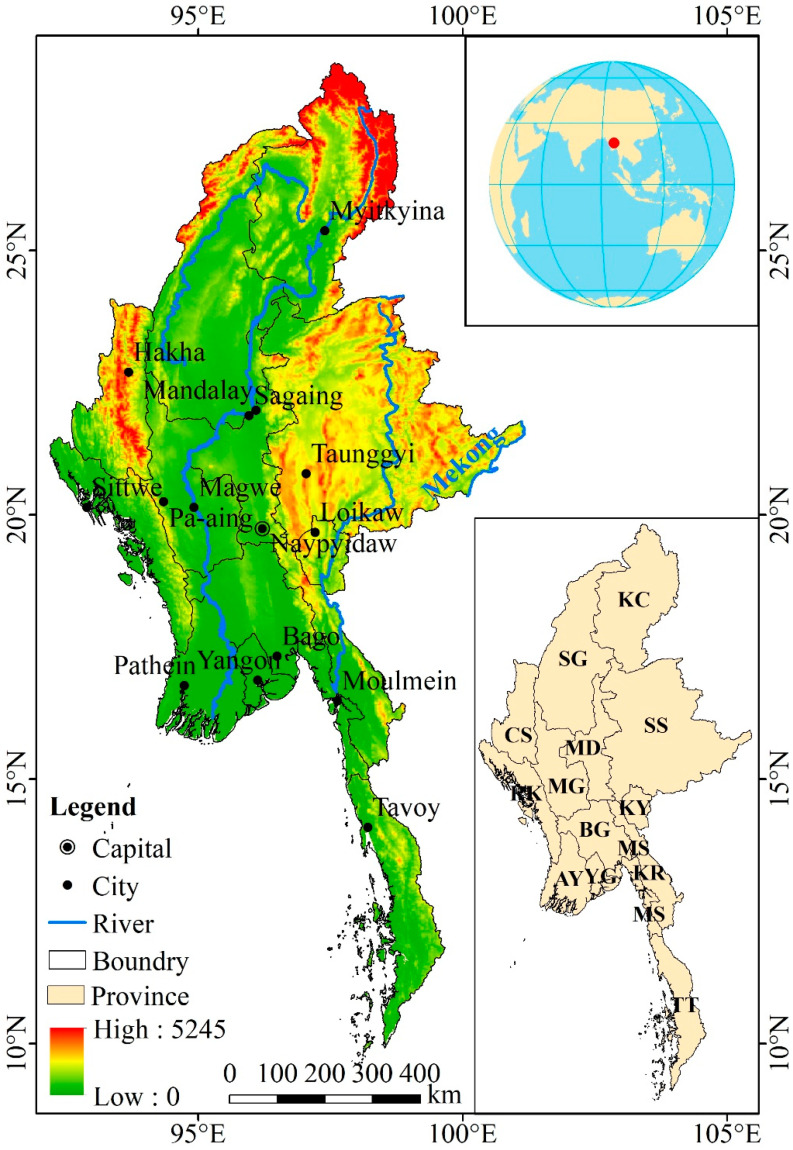
Location and topography map of Myanmar. AY: Ayeyarwady; BG: Bago; CS: Chin State; KC: Kachin State; KR: Karen State: KY: Kayah State; MD: Mandalay; MG: Magway; MS: Mon State; RK: Rakhine; SG: Sagaing; SS: Shan State; TT: Tanintharyi; and YG: Yangon.

**Figure 2 ijerph-20-02409-f002:**
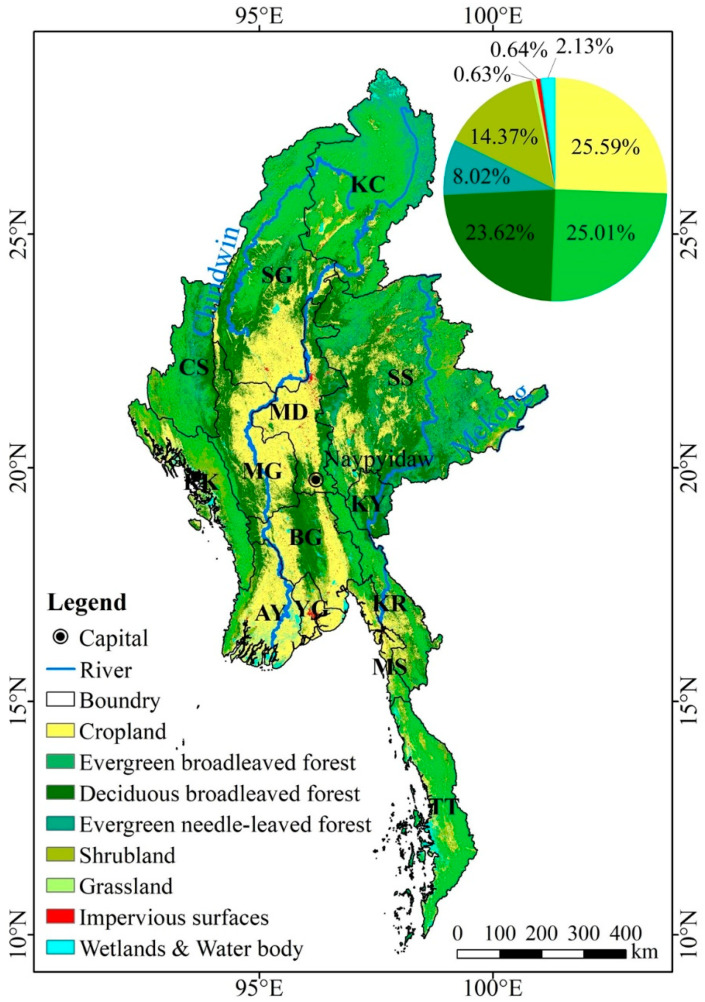
LUC map of Myanmar in 2020.

**Figure 3 ijerph-20-02409-f003:**
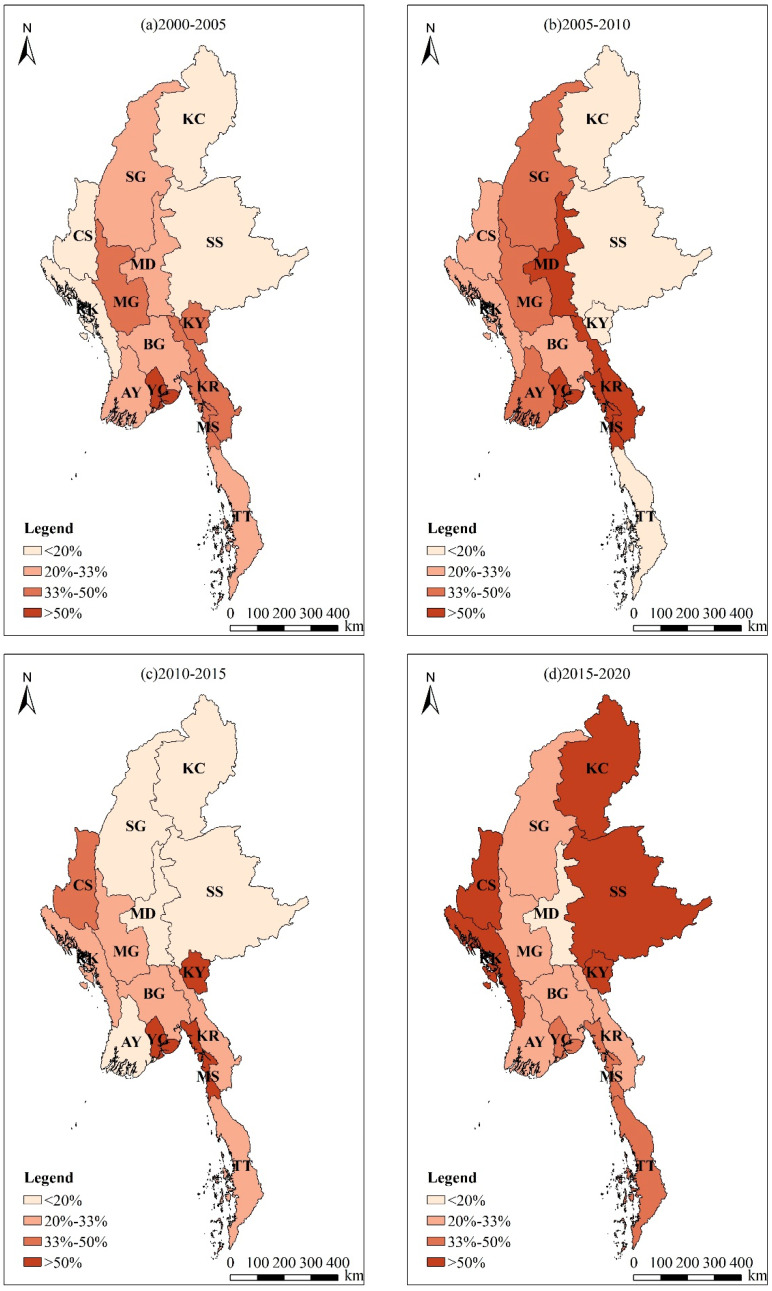
Change tendencies of LUC types of Myanmar during 2000–2020.

**Figure 4 ijerph-20-02409-f004:**
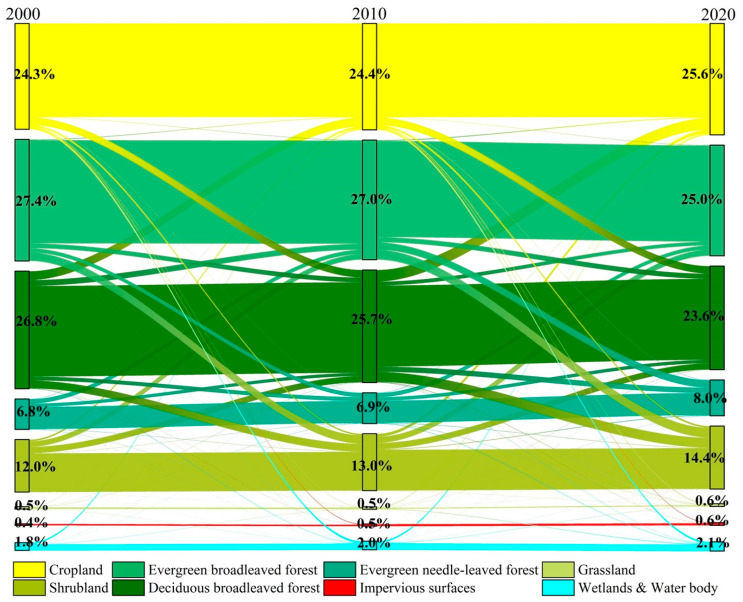
Conversion between different LUC types in Myanmar from 2000 to 2020.

**Figure 5 ijerph-20-02409-f005:**
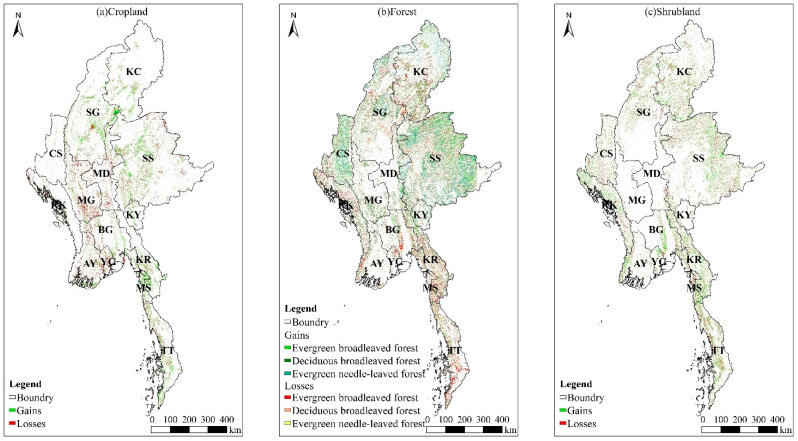
Spatial distributions of different LUC gains/losses from 2000 to 2020.

**Figure 6 ijerph-20-02409-f006:**
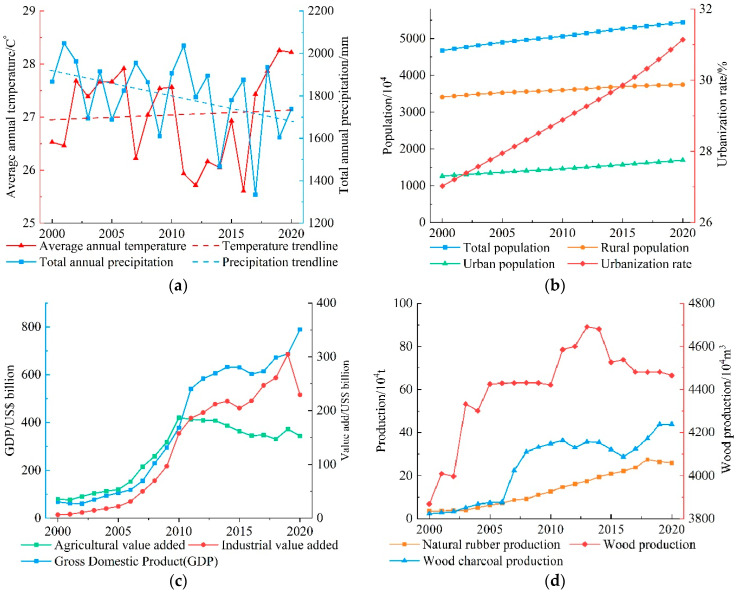
Development status in Myanmar from 2000 to 2020. (**a**) Climate factors; (**b**) demographic factors; (**c**) economic factors; (**d**) forestry factors; and (**e**) agricultural factors.

**Table 1 ijerph-20-02409-t001:** LULC classification system of GLC_FCS30.

Code	Level 1 Classes	LC ID	Level 2 Classes
1	Cropland	10	Rainfed cropland
		11	Herbaceous cover
		12	Tree or shrub cover (orchard)
		20	Irrigated cropland
2	Forest	51	Open evergreen broad-leaved forest
		52	Closed evergreen broad-leaved forest
		61	Open deciduous broad-leaved forest (0.15 < fc < 0.4)
		62	Closed deciduous broad-leaved forest (fc > 0.4)
		71	Open evergreen needle-leaved forest (0.15 < fc < 0.4)
		72	Closed evergreen needle-leaved forest (fc > 0.4)
		81	Open deciduous needle-leaved forest (0.15 < fc < 0.4)
		82	Closed deciduous needle-leaved forest (fc > 0.4)
		91	Open mixed-leaf forest (broad-leaved and needle-leaved)
		92	Closed mixed-leaf forest (broad-leaved and needle-leaved)
3	Shrubland	120	Shrubland
		121	Evergreen shrubland
		122	deciduous shrubland
4	Grassland	130	Grassland
5	Wetlands	180	Wetlands
6	Impervious surfaces	190	Impervious surfaces
7	Bare areas	140	Lichens and mosses
		150	Sparse vegetation (fc < 0.15)
		152	Sparse shrubland (fc < 0.15)
		153	Sparse herbaceous (fc < 0.15)
		200	Bare areas
		201	Consolidated bare areas
		202	Unconsolidated bare areas
8	Water body	210	Water body
9	Permanent ice and snow	220	Permanent ice and snow
		250	Filled value

**Table 2 ijerph-20-02409-t002:** LULC classification system in Myanmar.

Code	Level 1 Classes	LC ID	Level 2 Classes
1	Cropland	10	Rainfed cropland
		11	Herbaceous cover
		12	Tree or shrub cover (orchard)
		20	Irrigated cropland
2	Evergreen broad-leaved forest	51	Open evergreen broad-leaved forest
		52	Closed evergreen broad-leaved forest
3	Deciduous broad-leaved forest	61	Open deciduous broad-leaved forest (0.15 < fc < 0.4)
		62	Closed deciduous broad-leaved forest (fc > 0.4)
4	Evergreen needle-leaved forest	71	Open evergreen needle-leaved forest (0.15 < fc < 0.4)
		72	Closed evergreen needle-leaved forest (fc > 0.4)
5	Shrubland	120	Shrubland
		121	Evergreen shrubland
		122	deciduous shrubland
6	Grassland	130	Grassland
		150	Sparse vegetation (fc < 0.15)
		200	Bare areas
		201	Consolidated bare areas
		202	Unconsolidated bare areas
7	Impervious surfaces	190	Impervious surfaces
8	Wetlands & Water body	180	Wetlands
		210	Water body
		220	Permanent ice and snow

**Table 3 ijerph-20-02409-t003:** Driving factors system of LUCC in Myanmar.

Category	Index	unit
**Climate**	X1 Average annual temperature	°C
	X2 Total annual precipitation	mm
**Social development**	X3 Total population	10^4^
	X4 Rural population	10^4^
	X5 Urban population	10^4^
	X6 Urbanization rate	%
**Economic development**	X7 Gross Domestic Product (GDP)	USD billion (current USD)
	X8 Agricultural value added	USD billion (current USD)
	X9 Industrial value added	USD billion (current USD)
	X10 Natural rubber production	10^4^ t
	X11 Wood charcoal production	10^4^ t
	X12 Wood production	10^4^ m^3^
	X13 Vegetable production	10^4^ t
	X14 Oil crop production	10^4^ t
	X15 Food production	10^4^ t
	X16 Fruit production	10^4^ t

Note: X1–X2 are from GLDAS-2.1 and PERSIANN-CDR datasets; X3–X9 are from the World Bank; X10–X16 are from the World Food and Agriculture Organization.

**Table 4 ijerph-20-02409-t004:** Component matrix of the principal component analysis.

Variables	Description	Component		
		F1-Socio-Economic	F2-Climate	F3-Agriculture
X1	Average annual temperature	0.637	−0.356	−0.425
X2	Total annual precipitation	−0.203	0.881	0.362
X3	Total population	0.979	−0.178	0.097
X4	Rural population	0.986	−0.144	0.040
X5	Urban population	0.968	−0.204	0.142
X6	Urbanization rate	0.964	−0.206	0.157
X7	Gross Domestic Product (GDP)	0.945	−0.108	0.307
X8	Agricultural value added	0.884	0.467	0.007
X9	Industrial value added	0.969	0.104	0.223
X10	Natural rubber production	0.949	−0.171	0.263
X11	Wood charcoal production	0.966	0.155	0.121
X12	Wood production	0.839	−0.066	−0.496
X13	Vegetable production	0.956	0.132	−0.130
X14	Oil crop production	0.858	0.466	−0.181
X15	Food production	0.664	0.489	−0.563
X16	Fruit production	0.989	−0.077	0.111
Variance (%)		56.29%	27.14%	13.46%
Eigenvalues		9.00	4.34	2.15

**Table 5 ijerph-20-02409-t005:** The relationship between the area of different LUC types and the three principal components.

Cropland	Y1=188,745.6***+4258.8× F1**+1089.7×F3*	R^2^ = 0.99, *p* < 0.01
Evergreen broad-leaved forest	Y2=202,771.6***−7311.9× F1*	R^2^ = 0.90, *p* < 0.05
Deciduous broad-leaved forest	Y3=193,633.8***−8608.8×F1*	R^2^ = 0.84, *p* < 0.05
Shrubland	Y4=100,310.6***+6547.5×F1*	R^2^ = 0.82, *p* < 0.05
Grassland	Y5=4162.1***+546.7×F1**	R^2^ = 0.97, *p* < 0.01
Impervious surfaces	Y6=3842.3***+722.5×F1*	R^2^ = 0.79, *p* < 0.05

Note: * indicates passing the test of *p* = 0.05, ** indicates passing the test of *p* = 0.01, and *** indicates passing the test of *p* = 0.001.

**Table 6 ijerph-20-02409-t006:** Results of multiple stepwise regression analysis of the area and driving factors for different LUC types.

Cropland	Y1=109,915.1**+3037.2×X6**−2.8×X15*	R^2^ = 0.99, *p* < 0.01
Evergreen broad-leaved forest	Y2=335,789.6***−4591.4×X6*	R^2^ = 0.91, *p* < 0.05
Deciduous broad-leaved forest	Y3=468,415.7***−10454.4×X6***	R^2^ = 0.98, *p* < 0.001
Shrubland	Y4=37,508.0***+42.7×X5***	R^2^ = 0.99, *p* < 0.001
Grassland	Y5=4817.4**+6.3×X9**−0.4×X15*	R^2^ = 0.99, *p* < 0.05
Impervious surfaces	Y6=−9808.8***+2.7×X3***	R^2^ = 0.99, *p* < 0.001

Note: * indicates passing the test of *p* = 0.05, ** indicates passing the test of *p* = 0.01, and *** indicates passing the test of *p* = 0.001.

## Data Availability

Not applicable.

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
