# Peer review of "Myanmar’s Land Cover Change and Its Driving Factors during 2000–2020"

_ijerph, 2023, doi:10.3390/ijerph20032409_

Round 1
Reviewer 1 Report
Comments to Authors:
1. Change the title. Make it simple and unique.
2. In the abstract, the results are subdivided into 4 different parts. However, it is not clear how these parts are related to the problem statement and methods used to achieve objectives. Hence, the authors are advised to revisit the abstract and make necessary changes in this regard.
3. From the abstract, one cannot measure the scope of the study. Make certain corrections.
4. Novelty, research contribution, and future direction be added to the abstract in certain parts of the manuscript. Correction is required.
5. Revisit keywords. See the rules to provide keywords in the manuscript.
6. The introduction section ended with research questions. This is not a good approach. Add the remainder of the manuscript, contribution, and applications at the end of this section.
7. Show and highlight facts and figures in more detail to enrich the introduction section.
8. Add literature as a review section. Include related case studies, mathematical models, facts, and figures. This should clearly highlight the global instance concerning the study theme.
9. In the end, clarify the research gap, that authors want to abridge. Correction is required.
10. Add suitable literature that should link to the methodology section. This is an important correction.
11. The methodology is stand-alone and vague. No link is found between the methods used and the objectives of the study. Make certain changes to establish a link.
12. Results should satisfy the objective of the study. No link is found between the study findings, methods, objectives, and problems of the study.
13. A futuristic approach or parameter is missing. Authors are advised to predict the future scenario up to the year 2050. Make certain corrections in all sections of the manuscript in this aspect.
14. Add more maps in the result sections accordingly. Corrections are required.
15. Future directions, policy implications, and research contributions are also missing.
16. A conclusion should satisfy all the comments given in this report. Rewrite the conclusion section.
17. Verify the scope and limitations of the study.
18. The English of the manuscript is so complex. Proofread and make it simple. Establish a flow between paragraphs and subheadings.
Authors are advised to rewrite the whole article. Major modifications and changes are required. Add suitable literature and establish a link between various sections of the manuscript.
Author Response
Thank you for your careful review. Your questions and suggestions are very helpful to improve the quality of the article. We have carefully checked the questions you raised. Please find your original comment (black) and our reply (blue) in the uploaded Word. All line numbers refer to those in the revised version.

Reviewer 2 Report
This paper focused on spatial Pattern Change of Land Use/Cover and Driving Factors in Myanmar during 2000-2020,which are important for regional ecological protection and sustainable development. In addition, the authors used GLC_FCS30 datasets and socio-economic data to analyze spatial pattern and driving mechanism of LUCC in Myanmar. The objectives of this study are not appropriate for this journal and there are still some major Issues to further illustrate the manuscript.
1. There are too many studies on the analysis of land use change and its drivers. This article does not clearly identify the nature of scientific problem, and not innovative. The data sources are not innovative, the research methods are not innovative, it is just a data accumulation and simple analysis, which is meaningless.
2. In response to the three questions raised in the introduction section, the first question is whether there is an error in the format of the writing.
3. In the data and methods section, many data that are well irrelevant to this paper are put into the manuscript, which is just a mere accumulation of data and meaningless. The methodology used is not innovative and not meaningful to the study.
4. The manuscript only introduces the basic principles of principal component analysis and multiple regression analysis, and how they are specifically used in this study is not introduced
5. Is there any relationship between warming and rainfall reduction on land use type? 1700mm annual rainfall belongs to tropical and subtropical climate and will not have any effect on LUCC change. The second principal component did not have any relationship with all land use changes.
6. Principal component analysis has been carried out in manuscript, why do we need to establish the relationship between the original variables and LUCC? What is the point of doing principal component analysis? The whole paper is written without logic, it is just a pile of data.
7. There many grammatical spelling errors in the manuscript, for example, LUCC is written as LUC, check the MS, be sure the language should be improved.
Author Response
Thank you for your review. Our research objective is to study the land use change and its driving factors in Myanmar from 2000 to 2020. It is consistent with the environmental theme of this magazine. Your questions and suggestions are very helpful to improve the quality of the article. We have carefully checked the questions you raised. Please find your original comment (black) and our reply (blue) in the uploaded Word. All line numbers refer to those in the revised version.

Round 2
Reviewer 1 Report
The authors still not followed the review report 1. for example:
1. Authors were asked to include review section and take out some literature to put in this new section. Authors were also advised to include fresh literature that support methodology and filled research gap. No compliance made.
2. Future situation is still missing about the land-use changes in Myanmar. Authors were asked to predict. No compliance made.
3. Methodology section is still stands alone. Establish a link between review and methods.
4. GIS-based maps should be colorful. High resolution color images should be included instead of black & white.
5. Figures should show different land use patterns highlighted with different color schemes.
6. Proper legends be given in all GIS-based Figures. Correct it.
7. Include Literature Table that should clarify related case studies.
8. Slash should be avoided in the Title.
Author Response
Thank you for such a careful review. The questions and suggestions you raised last time were very helpful in improving the quality of the article. We have carefully checked these issues you raised this time. Please find your original comments (in black) and our responses (in blue) in the uploaded Word. All line numbers refer to the line numbers in the revised version.

Reviewer 2 Report
I am satisfied with the answer given by the author, but there are still some major issues to further illustrate the manuscript
1. English expression of the manuscript are very poor , for example, line15,110,there are still errors, and many of the sentences are Chinese English, so please find a relevant retouching company to improve the quality of themanuscript.
2. Why only forest land is classified at the Tier 2 and all others are classified at the Tier 1, and the accuracy of the Tier 2 classification is only 68.7%,
3. In the manuscript, the transfer matrix is calculated in ArcGIS 10.2 software, how to use the markov model? I know the Markov model is used for prediction, please describe in detail the steps to build the transfer matrix using Markov model in this paper
Author Response

(The authors gave the same response as above.)
